# Sleep Duration and Sleep Quality Associated with Dietary Index in Free-Living Very Old Adults

**DOI:** 10.3390/nu10111748

**Published:** 2018-11-13

**Authors:** Lovro Štefan, Ivan Radman, Hrvoje Podnar, Goran Vrgoč

**Affiliations:** 1Faculty of Kinesiology, University of Zagreb, 10 000 Zagreb, Croatia; ivan.radman@kif.hr (I.R.); hrvoje.podnar@kif.hr (H.P.); 2Department of Orthopedics, General Medical Hospital ‘Sveti Duh’, 10 000 Zagreb, Croatia; gvrgoc@gmail.com

**Keywords:** sleep hygiene, geriatrics, nutrition, association, logistic regression

## Abstract

The main purpose of the present study was to determine the associations between sleep duration and sleep quality with respect to dietary habits. In this cross-sectional study, 810 free-living older adults aged ≥85 years were recruited from six neighborhoods from the city of Zagreb. Diet, sleep duration and sleep quality were assessed using self-reported questionnaires. The associations between sleep duration and sleep quality with respect to dietary habits were examined using generalized estimating equations with Poisson regression analyses. After adjusting for body-mass index, self-rated health, psychological distress, physical activity, socioeconomic status, chronic disease/s, sex and age, we revealed that ‘short’ (<7 h; Rate Ratio (RR) = 0.43; 95% CI(confident interval) 0.30 to 0.64) and ‘long’ (>8 h; RR = 0.26; 95% CI 0.11 to 0.48) sleep durations and ‘good’ sleep quality (RR = 1.13; 95% CI 1.06 to 1.20) were associated with a ‘moderate-to-high’ healthy diet. When sleep duration and sleep quality were entered simultaneously into Model 3, ‘short’ (RR = 0.28; 95% CI 0.16 to 0.44 and ‘long’ (RR = 0.27; 95% CI 0.15 to 0.52) sleep duration and ‘good’ sleep quality (RR = 1.14; 95% CI 1.05 to 1.25) remained associated with a ‘moderate-to-high’ healthy diet. Our study shows that both ‘short’ and ‘long’ sleep duration and ‘good’ sleep quality are associated with ‘moderate-to-high’ healthy diets.

## 1. Introduction

In the last 15 years, a great deal of attention has been given to sleep duration and its association with health in older adults [1,2]. Specifically, both ‘short’ (<7 h) and ‘long’ (>8 h) sleep durations have been associated with negative health outcomes, such as higher mortality rates [3], and cardiovascular [4], metabolic [5] and mental diseases [6].

Among numerous factors associated with sleep [7], dietary patterns have modulated both sleep duration and sleep quality. For instance, evidence shows that sleeping for 7–8 h is positively associated with better diet quality [8], which is reflected by higher protein, vegetables and fruit consumption [9] and lower fat intake [9]. Similar to sleep duration, studies have shown that ‘good’ sleep quality is associated with higher vegetable intake and is negatively associated with fat intake [10]. However, the causality of the association remains unclear. A recent longitudinal study trying to explore the associations between adherence to the Mediterranean diet, sleep duration and sleep quality over a median follow-up of 2.8 years showed that a higher adherence to the Mediterranean diet was associated with a lower risk of sleep duration change and with better sleep quality in older adults [11]. From a physiological point of view, the dietary quantity and quality of specific nutrients may impact regulatory hormonal pathways to alter sleep [12]. 

The evidence about the aforementioned associations in Croatia is scarce. Although not conducted among the elderly population, one recent Croatian study among the general population has shown that most respondents report consuming one serving of vegetables and one piece of fruit a day and whole grains every other day, yet 50% of the respondents reported consuming fast food in a week, pointing out that their diet quality is below international dietary recommendations [13]. Unfortunately, we found no study exploring the prevalence of sleep duration and sleep quality and their associations with dietary habits in Croatian older adults. 

Therefore, the main purpose of the present study was to determine the associations between sleep duration and sleep quality with respect to dietary habits in a relatively large sample of Croatian older adults.

## 2. Materials and Methods

### 2.1. Study Participants

In this cross-sectional study, we recruited participants aged ≥85 years from six conveniently selected neighborhoods in the city of Zagreb. At the second stage, we spread the information about the study aims and hypothesis via flyers and posters in each neighborhood. We also contacted local nursing homes to help us organize the procedure. The inclusion criteria were: (1) age ≥85, (2) free of cognitive diseases, (3) able to read and write without assistance. The measuring protocol lasted from July to October 2018 and we approached each neighborhood only once. We put the date of the measurement on fliers as well as other information, such as researcher telephone numbers and e-mail addresses. According to the National Bureau of Statistics [14], in 2017 there were 4523 and 11,023 men and women aged ≥85 years in the city of Zagreb, respectively. In order to have 90% confidence in detecting a difference in proportions of 0.15 for meeting physical activity (PA) guidelines between any two groups, where the sleep duration or sleep quality of each group comprises one fifth of the sample, under the worst case that the overall proportion is 0.5 and using a two-sided test at the 0.05 level with 10 predictors, a total of 783 participants would be needed to produce usable data. In order to correct for a possibly large drop-out rate, at the beginning we recruited 1040 individuals (out of 1440; 72.2%) who met the inclusion criteria. However, 125 of them did not provide full data for diet and sleep variables and 105 of them did not want to participate and were additionally excluded from further analyses. All procedures performed in this study were anonymous and in accordance with the Declaration of Helsinki. The Institutional Review Board of the Faculty of Kinesiology approved the study (Ethics code: 10/7/2018). Before the study, each participant had given their written informed consent to participate in the study.

### 2.2. Dietary Habits (Outcome Variable)

To assess dietary habits, we used the Elderly Dietary Index score [15]. The score is constructed using 10 questions about the consumption frequency of meat, fish, fruits, vegetables, grains, legumes, olive oil and alcohol, as well as the type of bread and dairy products that the individual consumes. Scores from 1 to 4 were assigned to all components of the index. The Elderly Dietary Index total score had a range between 10 and 40 [15]. Higher values of the Elderly Dietary Index indicate a greater adherence to dietary recommendations for older adults or otherwise a greater adherence to a healthful dietary pattern. Participants were divided into three groups as follows: (1) Those considered as having an unhealthy diet or a diet away from recommendations (i.e., an Elderly Dietary Index total score between 10 and 28), (2) those considered as having a moderate healthy diet or a diet close to recommendations (i.e., an Elderly Dietary Index total score between 29 and 31) and (3) those considered as having a healthy diet or a diet very close to recommendations (i.e., an Elderly Dietary Index total score ≥32). For the purpose of this study, we dichotomized the outcome, where participants with less than 29 points were considered to have a ‘low’ healthy diet, compared to those scoring ≥29 points that were considered to have a ‘moderate-to-high’ healthy diet.

### 2.3. Sleep Duration and Sleep Quality

Sleep duration was assessed by asking participants the following question: ‘On average, how many hours of sleep do you get in a 24h period?’ [4] The response was a numerical variable. Finally, we grouped the participants into ‘short’ (<7 h), ‘normal’ (7–8 h) and ‘long’ sleepers.

To assess sleep quality, we asked the participants the following question: ‘How would you rate your sleep quality?’ with four answers: (1) Very poor, (2) poor, (3) good and (4) very good. For the purpose of this study, we dichotomized the outcome as ‘poor’ (very poor and poor) vs. ‘good’ (good and very good) sleep quality.

### 2.4. Covariates

The height and weight of the participants was self-reported. From that information, we calculated the body-mass index of each participant using the following formula: Weight (kg)/height^2^ (m^2^). Thus, we divided the participants into normal (<25 kg/m^2^) and overweight/obese (≥25 kg/m^2^) groups. Self-rated health was assessed using a one-item question: ‘How would you rate your health?’ Answers were arranged as follows: (1) Very poor, (2) poor, (3) fair, (4) good and (5) excellent. We dichotomized the outcome variable into ‘good’ (good and excellent) vs. ‘poor’ (poor, very poor and fair) self-rated health. Psychological distress was assessed using Kessler’s six-item questionnaire: (1) ‘How often during the past 30 days did you feel nervous?’, (2) ‘How often during the past 30 days did you feel hopeless?’, (3) ‘How often during the past 30 days did you feel restless or fidgety?’, (4) ‘How often during the past 30 days did you feel so depressed that nothing could cheer you up?’, (5) ‘How often during the past 30 days did you feel that everything was an effort?’ and (6) ‘How often during the past 30 days did you feel worthless?’ [16] Each question is scored from 0 (none of the time) to 4 (all the time). The scores of each question are summed up between 0 and 24, with a lower score indicating a lower level of psychological distress. Kessler et al. [16] showed that responses with <13 points vs. ≥13 points discriminated participants with and without psychological distress. Socioeconomic status was assessed by a one-item question: ‘How would you perceive your socioeconomic status?’ The three options were arranged and then collapsed into two categories. To assess physical activity in the last 7 days, we used the adapted version of the International Physical Activity Questionnaire-short form, a reliable and valid instrument designed to measure physical activity in respondents aged ≥65 [17]. The questionnaire provides information about the time and number of days spent in light, moderate and vigorous intensity physical activity. For each participant, we calculated the time spent in moderate and vigorous physical activity. According to the World Health Organization [18], elderly aged ≥65 should participate in ‘at least 150 min of moderate-intensity aerobic PA throughout the week or do at least 75 min of vigorous-intensity aerobic physical activity throughout the week or an equivalent combination of both’. Thus, we categorized the participants who met the aforementioned recommendation as ‘sufficiently’ active compared with the participants who did not meet the recommended levels as ‘insufficiently’ active. The presence or absence of a chronic disease was asked by the one-item question: ‘Have you ever been told by a doctor, that you suffer from any kind of chronic disease?’ with ‘yes’ and ‘no’ answers.

### 2.5. Data Analysis

The basic descriptive statistics of the study participants are presented as frequencies (n) and percentages (%). We used logistic regression analysis to calculate odds ratios with a 95% Confidence interval (CI) (odds ratio (OR); 95% CI) in Table 1. To examine the associations of sleep duration and sleep quality with the Elderly Diet Index we used generalized estimating equations to model a ‘moderate-to-high’ healthy diet as a binary outcome using Poisson regression analysis with an exchangeable matrix with neighborhood clustering. We used residual analysis as the main diagnostics to check for variation in the data that cannot be explained by the model. In Model 1, we examined the association between sleep duration (using 7–8 h as a reference value) and a ‘moderate-to-high’ healthy diet. In Model 2, we examined the association between sleep quality (using ‘poor’ as a reference value) and a ‘moderate-to-high’ healthy diet. Finally, we examined sleep duration and sleep quality simultaneously in Model 3 to examine their associations with a ‘moderate-to-high’ healthy diet. All three models were adjusted for body-mass index, self-rated health, psychological distress, physical activity, socioeconomic status, chronic diseases, sex and age. Significance was set up at α = 0.05 and it was two-sided. All the analyses were performed in Statistical Package for Social Sciences Software, V.22 (IBM development company, Armonk, New York, NY, USA).

## 3. Results

Our sample was based on 810 older adults (response rate 78%; 16.0% men; total sample mean age ± SD: overall 87.60 ± 2.44 years; men 87.60 ± 2.30 years and women 87.70 ± 2.46; total sample mean height ± SD: overall 1.63 ± 0.09 m, men 1.68 ± 1.12 m and women 1.60 ± 0.92 m; total sample mean weight ± SD: overall 71.60 ± 13.14 kg, men 74.54 ± 12.60 kg and women 67.56 ± 14.28 kg; total sample mean body-mass index ± SD: overall 26.88 ± 4.34 kg/m^2^, men 26.43 ± 3.94 kg/m^2^ and women 26.39 ± 4.53 kg/m^2^). We analyzed the differences between those who had missing values and those who were included in the analyses. However, we found no significant differences between those two groups in terms of age (*p* = 0.265), body-mass index (*p* = 0.768), self-rated health (*p* = 0.318) and psychological distress (*p* = 0.437). The interaction effects between sex and diet (*p* = 0.123), sleep duration (*p* = 0.426) and sleep quality (*p* = 0.295) were non-significant, so we dropped the sex-stratified analysis.

Basic descriptive statistics of the study participants are presented in Table 1. From the total sample, almost 2/3 of the participants reported sleeping <7 h, while 88 participants were ‘long’ sleepers. More participants categorized as having ‘moderate-to-high’ healthy diets reported having ‘poor’ sleep quality. Also, more participants with ‘optimal’ sleep duration and ‘good’ sleep quality were categorized as having ‘moderate-to-high’ healthy diets.

Table 2 represents the associations between sleep duration (Model 1), sleep quality (Model 2) and both sleep duration and sleep quality (Model 3) with ‘moderate-to-high’ healthy diets. After adjusting for body-mass index, self-rated health, psychological distress, physical activity, socioeconomic status, chronic disease/s, sex and age, ‘short’ (<7 h; RR = 0.43; 95% CI 0.30 to 0.64) and ‘long’ (>8 h; RR = 0.26; 95% CI 0.11 to 0.48) sleep duration and ‘good’ sleep quality (RR = 1.13; 95% CI 1.06 to 1.20) were associated with ‘moderate-to-high’ healthy diets. When sleep duration and sleep quality were both analyzed in Model 3, ‘short’ (RR = 0.28; 95% CI 0.16 to 0.44 and ‘long’ (RR = 0.27; 95% CI 0.15 to 0.52) sleep duration and ‘good’ sleep quality (RR = 1.14; 95% CI 1.05 to 1.25) remained associated with a ‘moderate-to-high’ healthy diet. Of note, when we calculated RRs and 95% CI for the association between sleep duration and sleep quality with diet unadjusted for potential confounders, we found similar values for ‘short’ (RR = 0.27; 95% CI 0.18 to 0.36; *p* < 0.001) and ‘long’ (RR = 0.28; 95% CI 0.14 to 0.58; *p* < 0.001) sleep durations and ‘good’ sleep quality (RR = 1.17; 95% CI 1.09 to 1.26; *p* < 0.001). 

## 4. Discussion

The main purpose of the present study was to determine the associations between sleep duration and sleep quality with respect to dietary habits. This study shows that both ‘short’ and ‘long’ sleep durations are negatively associated, and ‘good’ sleep quality is positively associated with a ‘moderate-to-high’ healthy diet when entered both separately and simultaneously into our models.

Our results are in accordance with previous cross-sectional [8,10,17] and longitudinal studies [11]. Specifically, a lower consumption of proteins [8], fibers [10], fruits and vegetables [8] and unhealthy dietary habits, such as skipping breakfast [19], have been associated with both ‘short’ and ‘long’ sleep durations. To determine the causal association, a longitudinal study by Campanini et al. [11] showed that a higher adherence to the Mediterranean diet was associated with a lower risk of change in sleep duration and with better sleep quality in older adults. As mentioned in the ‘Introduction’ section, regarding the association between sleep duration and dietary habits, specific nutrients may be effective triggers to leverage ‘good’ sleep throughout many biological mechanisms [12].

As for sleep duration, our study showed that ‘good’ self-reported sleep quality was positively associated with ‘moderate-to-high’ healthy diets. Our findings are in accordance with previous studies which showed that a low protein, fruit and vegetable intake and a higher intake of fat were both associated with ‘poor’ sleep quality [10,20,21]. 

Both ‘short’ and ‘long’ sleep durations and sleep quality are associated with diet. However, the causality of such association still remains unclear. A study by Frank et al. [12] tried to explain public health and the clinical implications of diet and sleep physiology and how these associations were related to chronic diseases. Their review article showed that extremes of sleep duration change sleep patterns, hormonal levels and circadian rhythm, which additionally led to weight-related problems (associated with obesity and metabolic syndrome) and cardiovascular diseases. Thus, optimal sleep and healthy diet may be mediators towards health status, especially for older adults. 

The National Sleep Foundation [22] reported that diet and sleep quality were both very poor among the U.S. population and this trend has been continuing to the present day. Moreover, protective lifestyle factors, such as physical activity and eating ‘healthy diets’, which might be affected by sleep and could be effective against chronic diseases decreased during1988 to 2006, pointing out that strategies aiming to enhance food quality and food intake, sleep duration and sleep quality should be implemented, especially within the school system, since the habits that children adopt at this age persist later in life. 

Our study has several limitations. Due to a cross-sectional design, the associations between sleep duration and sleep quality with respect to diet must be interpreted with caution. It is possible that ‘poor’ food choice led to both ‘short’ and ‘long’ sleep durations and ‘poor’ sleep quality. Secondly, we used self-reported measures. It is possible that individuals misreported their sleep durations, sleep quality levels and dietary intake, which could lead to considerable measurement error, recall bias and social desirability effect. Also, the association between sleep quality and diet should be interpreted with caution (despite using a cross-sectional design), since sleep quality already incorporates some aspects of sleep duration. Finally, we did not adjust for some crucial covariates, such as obstructive sleep apnea and breathing problems, which might affect the aforementioned associations. Future studies should use more objective methods, like accelerometers with a later follow-up, in order to establish the causality of the association between sleep and diet.

## 5. Conclusions

In conclusion, our study shows that participants reporting ‘short’ and ‘long’ sleep durations are less likely to have a ‘moderate-to-high’ healthy diet and those participants reporting ‘good’ sleep quality were more likely to have a ‘moderate-to-high’ healthy diet after adjusting for several covariates. However, our results should be interpreted with caution, due to a cross-sectional design and the possible overlapping effect between the sleep duration and sleep quality variables. Nevertheless, by improving sleep duration (i.e., having an earlier bed time), it is possible that dietary patterns would change, yet this association could also go in other direction.

## Figures and Tables

**Table 1 nutrients-10-01748-t001:** Basic descriptive statistics of the study participants, Croatia (*N* = 810).

Study Variables	Total (*N* = 810)	‘Low’ Healthy Diet (*N* = 666)	‘Moderate-to-High’ Healthy Diet (*N* = 144)	OR (95% CI; *p*-Value)
	*N* (%)	*N* (%)	*N* (%)	
**Sleep duration**				
Short (<7 h)	498 (61.5)	434 (65.2)	64 (44.4)	0.39 (0.26 to 0.64; *p* < 0.001)
Optimal (7–8 h)	224 (27.7)	154 (23.1)	70 (48.6)	Ref.
Long (>8 h)	88 (10.9)	78 (11.7)	10 (6.9)	0.26 (0.11 to 0.48; *p* < 0.001)
**Sleep quality**				
Poor	508 (62.7)	468 (70.3)	40 (27.8)	Ref.
Good	302 (37.3)	198 (29.7)	104 (72.2)	3.92 (2.44 to 6.31; *p* < 0.001)
**Body-mass index**				
Overweight/obesity	512 (63.2)	444 (66.7)	68 (47.2)	Ref.
Normal	298 (36.8)	222 (33.3)	76 (52.8)	1.60 (1.25 to 2.00; *p* = 0.003)
**Self-rated health**				
Poor	414 (51.1)	364 (54.6)	50 (34.7)	Ref.
Good	396 (48.9)	302 (45.4)	94 (65.3)	3.25 (1.95 to 5.39; *p* < 0.001)
**Psychological distress**				
High	200 (24.7)	186 (27.9)	14 (9.7)	Ref.
Low	610 (75.3)	480 (72.1)	130 (90.3)	3.15 (1.67 to 5.97; *p* < 0.001)
**Physical activity**				
Insufficiently active	630 (77.8)	562 (84.4)	70 (48.6)	Ref.
Sufficiently active	180 (21.2)	104 (15.6)	76 (52.8)	5.26 (3.31 to 8.36; *p* < 0.001)
**Socioeconomic status**				
Low	678 (83.7)	566 (85.0)	112 (77.8)	Ref.
Middle/high	132 (16.3)	100 (15.0)	32 (22.2)	1.35 (0.76 to 2.37; *p* = 0.287)
**Chronic disease/s**				
Yes	530 (65.4)	440 (66.1)	90 (62.5)	Ref.
No	280 (34.6)	226 (33.9)	54 (37.5)	0.95 (0.60 to 1.50; *p* = 0.876)
**Height (SD)**	1.63 (0.09)	1.63 (0.09)	1.63 (1.00)	1.00 (0.99 to 1.01; *p* = 0.978)
**Weight (SD)**	71.60 (13.14)	71.98 (12.76)	70.00 (14.64)	0.99 (0.97 to 1.01; *p* = 0.945)
**Sex**				
Men	132 (16.3)	106 (15.9)	26 (18.1)	Ref.
Women	678 (83.7)	560 (84.1)	118 (81.9)	1.01 (0.98 to 1.04; *p* = 0.869)
**Age (SD)**	87.60 (2.44)	87.52 (2.42)	87.90 (2.36)	1.01 (0.99 to 1.02; *p* = 0.912)

SD: Standard deviation; Ref.: Referent value; OR: odds ratio; CI: confident interval.

**Table 2 nutrients-10-01748-t002:** Rate ratios for ‘moderate-to-high’ healthy diet in the study participants, Croatia (*N* = 810).

Study Variables	Model 1	Model 2	Model 3
	RR (95% CI; *p*-Value)	RR (95% CI; *p*-Value)	RR (95% CI; *p*-Value)
**Sleep duration**			
Short (<7 h)	0.43 (0.30 to 0.64; *p* < 0.001)		0.28 (0.16 to 0.44; *p* < 0.001)
Optimal (7–8 h)	Ref.		Ref.
Long (>8 h)	0.26 (0.11 to 0.48; *p* < 0.001)		0.27 (0.15 to 0.52; *p* < 0.001)
**Sleep quality**			
Poor		Ref.	Ref.
Good		31.13 (1.06 to 1.20; *p* < 0.001)	1.14 (1.05 to 1.25; *p* < 0.001)
**Body-mass index**			
Overweight/obesity	Ref.	Ref.	Ref.
Normal	1.09 (1.03 to 1.15; *p* < 0.001)	1.09 (1.02to 1.16; *p* < 0.001)	1.08 (1.03 to 1.14; *p* = 0.002
**Self-rated health**			
Poor	Ref.	Ref.	Ref.
Good	1.06 (1.01 to 1.10; *p* < 0.001)	1.10 (1.02 to 1.20; *p* < 0.001)	1.08 (1.03 to 1.14; *p* < 0.001)
**Psychological distress**			
High	Ref.	Ref.	Ref.
Low	1.08 (1.06 to 1.10; *p* < 0.001)	1.11 (1.08 to 1.13; *p* < 0.001)	1.11 (1.09 to 1.12; *p* < 0.001)
**Physical activity**			
Insufficiently active	Ref.	Ref.	Ref.
Sufficiently active	1.24 (1.10 to 1.40; *p* < 0.001)	1.30 (1.15 to 1.48; *p* < 0.001)	1.20 (1.10 to1.28; *p* < 0.001)
**Socioeconomic status**			
Low	Ref.	Ref.	Ref.
Middle/high	1.05 (1.00 to 1.11; *p* = 0.050)	0.97 (0.92 to 1.03; *p* = 0.282)	0.97 (0.91 to 1.03; *p* = 0.312)
**Chronic disease/s**			
Yes	Ref.	Ref.	Ref.
No	1.01 (0.93 to 1.09; *p* = 0.850)	0.99 (0.95 to 1.04; *p* = 0.778)	1.00 (0.94 to 1.06; *p* = 0.980)
**Sex**			
Men	Ref.	Ref.	Ref.
Women	0.99 (0.95 to 1.03; *p* = 0.876)	1.01 (0.98 to 1.05; *p* = 0.884)	1.00 (0.99 to 1.02; *p* = 0.930)
**Age**	1.02 (0.99 to 1.05; *p* = 0.493)	1.04 (1.00 to 1.09; *p* = 0.050)	1.03 (0.98 to 1.08; *p* = 0.278)

RR: Rate Ratio; **Model 1:** Associations between sleep duration and ‘moderate-to-high’ healthy diet, results are adjusted for body-mass index, self-rated health, psychological distress, physical activity, socioeconomic status, chronic disease/s, sex and age. **Model 2:** Associations between sleep quality and ‘moderate-to-high’ healthy diet, results are adjusted for body-mass index, self-rated health, psychological distress, physical activity, socioeconomic status, chronic disease/s, sex and age. **Model 3:** Associations between sleep duration and sleep quality entered simultaneously into the model with a ‘moderate-to-high’ healthy diet, results are adjusted for body-mass index, self-rated health, psychological distress, physical activity, socioeconomic status, chronic disease/s, sex and age.

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
