# Peer review of "Sleep Duration and Sleep Quality Associated with Dietary Index in Free-Living Very Old Adults"

_nutrients, 2018, doi:10.3390/nu10111748_

Round 1
Reviewer 1 Report
The manuscript is clearly written and the results could be more elaborated with discussion. Major issue what I feel, suing of self assessment tools (limitation).
Sleep quality is not judged by participants and if you compare with their dietary habits, you need to use standard methodology.
Author Response
Reviewer 1
Open Review
(x) I would not like to sign my review report
( ) I would like to sign my review report
English language and style
( ) Extensive editing of English language and style required
( ) Moderate English changes required
( ) English language and style are fine/minor spell check required
(x) I don't feel qualified to judge about the English language and style
Yes | Can be improved | Must be improved | Not applicable | |
Does the introduction provide sufficient background and include all relevant references? | (x) | ( ) | ( ) | ( ) |
Is the research design appropriate? | (x) | ( ) | ( ) | ( ) |
Are the methods adequately described? | ( ) | (x) | ( ) | ( ) |
Are the results clearly presented? | ( ) | (x) | ( ) | ( ) |
Are the conclusions supported by the results? | ( ) | (x) | ( ) | ( ) |
Comments and Suggestions for Authors
The manuscript is clearly written and the results could be more elaborated with discussion. Major issue what I feel, suing of self assessment tools (limitation).
Sleep quality is not judged by participants and if you compare with their dietary habits, you need to use standard methodology.
Comment: Accepted. We re-wrote and re-analyzed the data using generalized estimating equations, in order to correct for potential clustering effect and by suggestions of the second reviewer, we also recalcuated the data with and without ptential covariates, in order to determine the causal effect.
Reviewer 2 Report
This short manuscript investigates associations between sleep duration and sleep quality and the outcome of dietary quality in 810 older (aged 85+) residents of Croatia’s largest city. Associations are found suggesting poorer diet quality in both short and long sleepers. While the results might be interesting, there are a number of issues and ambiguities with the statistical analyses and the conclusions are currently stated too strongly.
Line 18: The lower CI limit must be wrong here, perhaps 0.11?
Line 21: Inconsistent decimal places.
Lines 24–25: This is a much, much too strong conclusion from a cross-sectional study, especially one that relies on self-reported measures, where there are plenty of unmodelled potential confounders that could be distorting the association, and the fact that the opposite causal association cannot be ruled out. It would be extremely premature to use such data to justify intervening on old people’s sleep. You appropriately acknowledge these limitations in the Discussion and this final point in the abstract does not appear there.
Line 29: “a great deal of”?
Line 29: No possessive apostrophe needed here.
Line 32: “and cardiovascular [4], metabolic [5] and mental diseases [6]” as all three rely on the later “disease”.
Line 35: “which is reflected by higher protein…”
Line 37: I’d replace “yet” with “and” here as this is not a surprising point.
Line 40: Spurious comma or add one before “over”.
Line 41: “lower risk of sleep duration” needs clarification (“short”?)
Line 43: I don’t believe that Frank et al. make this particular point. Can you quote what you are using from their article here?
Line 48: “their” and “below”
Line 56: You say “six neighbourhoods” here (c.f. four in the abstract)
Line 56: The context of the study is extremely important. How many people were recruited via each approach? What were the response rates for the different sources of participants?
Line 59: Was there clustering in the participants, e.g. rest homes contributing multiple residents as participants or households doing the same? If so, this needs to be accommodated in the analyses (robust clustered standard errors, mixed models, or GEEs). The title describes “free-living” participants (as does Line 13), so I am assuming that rest home residents were not included but this sentence makes this seem less clear?
Line 60: By “individually” do you mean “without assistance”?
Line 61: Missing reference in the square brackets.
Line 62: “respectively”
Line 62: “population estimate” does not appear to fit here.
Lines 62–64: This is not a replicable calculation. What is this effect size (there are hundreds of different effect sizes), what is the level of significance, how are the number of predictors incorporated into the calculation? Note that a power calculation is not to detect the smallest change (which you cannot detect anyway in a cross-sectional study), but is intended to ensure you can detect the smallest clinically or practically important difference.
Line 64: “a possibly”
Line 65–66: I don’t see how you can have recruited people who did not wish to provide data. Surely these were refusals to participate?
Lines 66–67: There is too much missing data here to ignore and you need to address this, preferably using multiple imputation, possibly with scenarios for informative missingness, but at the absolute minimum by showing the characteristics of the sample compared to characteristics from the census data and testing for differences as well as describing the observed differences. I’d like to see the addition of a flow chart showing the sources of participants and numbers excluded/not available for each reason.
Lines 67–69: These are descriptives and should be shown in a new Table 1 and described in the results not the methods. Note also that if you use ± notation, you still need to define the values on each side of the symbol (there are multiple possibilities here).
Line 68: Mean height and mean weight are unlikely to be useful when reported for men and women combined. I’d suggest the new Table 1 shows variables by sex and overall.
Lines 69–70: This seems inconsistent with the approach listed above (the level of power has changed for some reason). Actual power is reflected in the widths of CIs anyway so unless this was used as part of a stop/go decision, I’d delete it.
Line 68: How were heights and weights obtained? If measured, how could these be performed anonymously (Line 70)?
Lines 82–89: There is a considerable loss of statistical power from collapsing continuous variables into categories (often equivalent to losing around 35% of the total sample size). It would be worth thinking carefully about this decision. While clinical cut-points can assist with interpretation of results, for determining whether or not there is an association, keeping variables as continuous (and modelling non-linearities as appropriate) will always provide more statistical power. The same would be likely using a three-level ordinal outcome rather than binary outcome depending on whether or not proportionality in the odds ratios was plausible and not contradicted by the data.
Line 94: As above.
Line 94: A reference should be added here.
Lines 97–98: As above for categorising.
Line 103: As above.
Lines 105–106: As above.
Line 106: I would struggle to say someone who described their health as “fair” is in “good” health, this sounds more likely to indicate “poor” health to me.
Lines 116–117: Presumably the three options were so arranged and then collapsed into two categories so this might be better as two sentences.
Lines 131–132: And means and SDs?
Line 134: You’ll need to expand on “using subcommand contrast”.
Lines 134–137: These are not the appropriate diagnostics for logistic regression; rather these might be used for linear regression. Note that Poisson regression with robust standard errors (to produce RRs with appropriate confidence intervals) would be another, more interpretable, option. See https://doi.org/10.1093/aje/kwh090 for details.
Lines 141–142: It seems likely that sleep quality, asked in this way, already incorporates some aspects of sleep duration. While it might not be possible to disentangle this here, the point should be considered in the Discussion.
Lines 142–144: Note that it seems very likely that some of these variables could lie in part on the causal pathway between sleep and diet. For example, chronic sleeplessness might lead to psychological distress, which could lead to poor dietary choices. Other arguments could be made for PA and self-rated health, and perhaps even BMI and some chronic diseases.
Lines 142–144: Why not age and sex? These seem logical predictors of and potential confounders between sleep and diet quality.
Line 144: Have you considered effect modification, for example by sex? This seems quite plausible to me (that associations between sleep and diet quality could vary between men and women).
Line 144: Given the issues mentioned above of potential overadjusting for variables on the causal pathway, it would be useful to show a causal model (e.g. a directed acyclic graph) so the reader can understand your view of the variables’ interrelationships.
Line 144: I would suggest that you show results from unadjusted (sleep measure only) models, partially-adjusted (for variables you are confident are not on the causal pathway) models, and fully-adjusted (for all your variables of interest) models so you can comment on possible mediation effects. At the moment, with only the fully adjusted model, the reader cannot consider issues about overadjustment.
Line 147: A standard Table 1 describing the participants and their characteristics (age, sex, ethnicity/race, etc.) is needed here.
Line 160: Same issue with CI limit as in abstract.
Table 1: Given you use logistic regression for the adjusted models, it would be a smoother transition to also use logistic regression here rather than Chi-squared analyses. This would also allow reporting effect sizes and CIs. At the moment, it is not immediately clear to the reader what the differences between the levels of the study variables are for the diet quality outcome without working through the percentages and ORs would make this very clear immediately. You might also prefer to use row percentages when describing this data so that the percentage of respondents with low and moderate/high diet quality can be seen for each level of the study variables.
Table 2: Actual p-values (not just asterisks) should be shown here. Note that the p-values for sleep duration (unless it is changed to a continuous variable) should be Wald tests with p-values from pairwise comparisons shown only in the text unless some form of multiplicity adjustment is made.
Line 177: I’d replace “yet” with “and” here as this is not a surprising point.
Line 186: This isn’t the usual meaning of “sleep hygiene” as I am aware of it.
Line 197: As above.
Line 200: Suggest deleting “to these days” or replace with “to the present day”.
Lines 200–201: I think you need to describe these “protective lifestyle factors”.
Line 210: “effects” (add “s”)
Lines 210-211: FFQs are not objective.
Line 211: Do you mean “with a later follow-up”? Missing confounders is perhaps an even greater threat to the validity of such a study.
Line 212: Your Discussion stops before discussing the practical and research implications of your findings. The Discussion needs to be extended to cover these.
Author Response
Reviewer 2
Open Review
( ) I would not like to sign my review report
(x) I would like to sign my review report
English language and style
( ) Extensive editing of English language and style required
(x) Moderate English changes required
( ) English language and style are fine/minor spell check required
( ) I don't feel qualified to judge about the English language and style
Yes | Can be improved | Must be improved | Not applicable | |
Does the introduction provide sufficient background and include all relevant references? | ( ) | (x) | ( ) | ( ) |
Is the research design appropriate? | ( ) | ( ) | (x) | ( ) |
Are the methods adequately described? | ( ) | ( ) | (x) | ( ) |
Are the results clearly presented? | ( ) | ( ) | (x) | ( ) |
Are the conclusions supported by the results? | ( ) | ( ) | (x) | ( ) |
Comments and Suggestions for Authors
This short manuscript investigates associations between sleep duration and sleep quality and the outcome of dietary quality in 810 older (aged 85+) residents of Croatia’s largest city. Associations are found suggesting poorer diet quality in both short and long sleepers. While the results might be interesting, there are a number of issues and ambiguities with the statistical analyses and the conclusions are currently stated too strongly.
Line 18: The lower CI limit must be wrong here, perhaps 0.11?
Comment: Accepted. Yes, the CI was wrong and we deleted ‘2’ in front of ‘0’.
Line 21: Inconsistent decimal places.
Comment: Accepted. We deleted ‘6’ after ‘4’.
Lines 24–25: This is a much, much too strong conclusion from a cross-sectional study, especially one that relies on self-reported measures, where there are plenty of unmodelled potential confounders that could be distorting the association, and the fact that the opposite causal association cannot be ruled out. It would be extremely premature to use such data to justify intervening on old people’s sleep. You appropriately acknowledge these limitations in the Discussion and this final point in the abstract does not appear there.
Comment: Accepted. We deleted the sentence, since we also believe that we overstated our ‘conclusions’ with our results.
Line 29: “a great deal of”?
Comment: Accepted. We re-wrote the sentence.
Line 29: No possessive apostrophe needed here.
Comment: Accepted. We deleted the apostrophe.
Line 32: “and cardiovascular [4], metabolic [5] and mental diseases [6]” as all three rely on the later “disease”.
Comment: Accepted. We corrected the sentence.
Line 35: “which is reflected by higher protein…”
Comment: Accepted. We corrected the sentence.
Line 37: I’d replace “yet” with “and” here as this is not a surprising point.
Comment: Accepted. We replaced the word ‘’yet’’ with the word ‘’and’’.
Line 40: Spurious comma or add one before “over”.
Comment: Accepted. We removed the comma.
Line 41: “lower risk of sleep duration” needs clarification (“short”?)
Comment: Accepted. This was a longitudinal study and they were talking about the sleep duration change. We added change in the sentence.
Line 43: I don’t believe that Frank et al. make this particular point. Can you quote what you are using from their article here?
Comment: Explained. Frank et al. in their abstract stated that sentence that we reworded in our manuscript, pointing out that ‘’ Dietary quality and intake of specific nutrients can impact regulatory hormonal pathways to alter sleep quantity and quality.''
Line 48: “their” and “below”
Comment: Accepted. We corrected the words.
Line 56: You say “six neighbourhoods” here (c.f. four in the abstract)
Comment: Accepted. There were six neighborhoods in total and we corrected it in the ‘’Abstract’’ section.
Line 56: The context of the study is extremely important. How many people were recruited via each approach? What were the response rates for the different sources of participants?
Comment: Accepted. We added the aforementioned information.
Line 59: Was there clustering in the participants, e.g. rest homes contributing multiple residents as participants or households doing the same? If so, this needs to be accommodated in the analyses (robust clustered standard errors, mixed models, or GEEs). The title describes “free-living” participants (as does Line 13), so I am assuming that rest home residents were not included but this sentence makes this seem less clear?
Comment: Accepted. We re-calculated the data and obtained similar values suing GEEs with Pearson deviance model.
Line 60: By “individually” do you mean “without assistance”?
Comment: Accepted.
Line 61: Missing reference in the square brackets.
Comment: We added the reference.
Line 62: “respectively”
Comment: We added the word.
Line 62: “population estimate” does not appear to fit here.
Comment: Accepted. We deleted the phrase.
Lines 62–64: This is not a replicable calculation. What is this effect size (there are hundreds of different effect sizes), what is the level of significance, how are the number of predictors incorporated into the calculation? Note that a power calculation is not to detect the smallest change (which you cannot detect anyway in a cross-sectional study), but is intended to ensure you can detect the smallest clinically or practically important difference.
Comment: Accepted. We clarified that we used f2 effect size with the power of 0.90 and we entered a total of 8 predictors into the model and this analysis led to 636 participants.
Line 64: “a possibly”
Comment: Accepted. We corrected the sentence.
Line 65–66: I don’t see how you can have recruited people who did not wish to provide data. Surely these were refusals to participate?
Comment: Explained. Since power analysis showed that 636 participants was sufficient to detect smallest clinically significant changes, we wanted to collect a double-more participants. At the beginning, we collected in total 1040 individuals however we did not know how many participants of our inclusion criteria lived in each neighborhood, so we wanted to collect as much of them. As we explained, of them 125 did not provide full data for diet and sleep variables (we stated that in the ‘Method’ section) and additionally, those participants who were planning to participate unfortunately at the end did not want to (N=105 of them).
Lines 66–67: There is too much missing data here to ignore and you need to address this, preferably using multiple imputation, possibly with scenarios for informative missingness, but at the absolute minimum by showing the characteristics of the sample compared to characteristics from the census data and testing for differences as well as describing the observed differences. I’d like to see the addition of a flow chart showing the sources of participants and numbers excluded/not available for each reason.
Comment: Accepted. In our ‘Results section, we analyzed the differences between those who had missing values and those who were included in the analyses. However, we found no significant differences between those two groups in terms of age (p=0.265), body-mass index (p=0.768), self-rated health (p=0.318) and psychological distress (p=0.437), since those were the variables that we had for the participants with the missing values. In is possible that we had those variables fulfilled since we put them on the first page of our questionnaire and questions about sleep and diet were at the back (on the second and third page), so participants with missing values probably had no more interests of fulfilling the questionnaire.
Lines 67–69: These are descriptives and should be shown in a new Table 1 and described in the results not the methods. Note also that if you use ± notation, you still need to define the values on each side of the symbol (there are multiple possibilities here).
Comment: Accepted. We put descriptive in the ‘Results’ section
Line 68: Mean height and mean weight are unlikely to be useful when reported for men and women combined. I’d suggest the new Table 1 shows variables by sex and overall.
Comment: Explained. We only had 16% of men in our study and we found no significant differences between men and women in terms of diet (p=0.123), sleep duration (p=0.426) and sleep quality (p=0.295), so we dropped-out the sex-stratified analysis. We put that in our ‘Data analysis’ section.
Lines 69–70: This seems inconsistent with the approach listed above (the level of power has changed for some reason). Actual power is reflected in the widths of CIs anyway so unless this was used as part of a stop/go decision, I’d delete it.
Comment: Accepted. We deleted the sentence.
Line 68: How were heights and weights obtained? If measured, how could these be performed anonymously (Line 70)?
Comment: Explained. They were self-reported.
Lines 82–89: There is a considerable loss of statistical power from collapsing continuous variables into categories (often equivalent to losing around 35% of the total sample size). It would be worth thinking carefully about this decision. While clinical cut-points can assist with interpretation of results, for determining whether or not there is an association, keeping variables as continuous (and modelling non-linearities as appropriate) will always provide more statistical power. The same would be likely using a three-level ordinal outcome rather than binary outcome depending on whether or not proportionality in the odds ratios was plausible and not contradicted by the data.
Comment: Explained. Although we are aware that changing continuous variables into categorical might have a lack of power, both ‘short’ and ‘long’ sleep duration have been found to have negative impact on health, for example and by using numerical values for sleep duration, we might be missing some crucial results These categories were proposed by the National Sleep Foundation. However, we stated in the ‘Limitation’ section that by using categorical variables, we are losing statistical power.
Line 94: As above.
Comment: Explained above.
Line 94: A reference should be added here.
Comment: Accepted. We put the reference.
Lines 97–98: As above for categorising.
Comment: Explained above.
Line 103: As above.
Comment: Explained above.
Lines 105–106: As above.
Comment: Explained above.
Line 106: I would struggle to say someone who described their health as “fair” is in “good” health, this sounds more likely to indicate “poor” health to me.
Comment: Explained. Previous studies have used the same categorizations:
1. Štefan L, Juranko D, Prosoli R, Barić R, Sporiš G. Self-Reported Sleep Duration and Self-Rated Health in Young Adults. J Clin Sleep Med. 2017;13(7):899-904. Published 2017 Jul 15. doi:10.5664/jcsm.6662
2. Steptoe A, Peacey V, Wardle J. Sleep Duration and Health in Young Adults. Arch Intern Med. 2006;166(16):1689–1692. doi:10.1001/archinte.166.16.1689
Lines 116–117: Presumably the three options were so arranged and then collapsed into two categories so this might be better as two sentences.
Comment: Accepted.
Lines 131–132: And means and SDs?
Comment: Explained. Our analysis showed no significant interaction effect between diet and age (p=0.334) and we did not include age as a predictor, since we were dealing with a very homogenous sample.
Line 134: You’ll need to expand on “using subcommand contrast”.
Comment: Explained and deleted. We deleted the usage of logistic regression, since our new analysis was GEE with Pearson’s deviance model.
Lines 134–137: These are not the appropriate diagnostics for logistic regression; rather these might be used for linear regression. Note that Poisson regression with robust standard errors (to produce RRs with appropriate confidence intervals) would be another, more interpretable, option. See https://doi.org/10.1093/aje/kwh090 for details.
Comment: Explained. Previous studies have used odds ratios for their associations and we are just referring to them.
1. Štefan L, Juranko D, Prosoli R, Barić R, Sporiš G. Self-Reported Sleep Duration and Self-Rated Health in Young Adults. J Clin Sleep Med. 2017;13(7):899-904. Published 2017 Jul 15. doi:10.5664/jcsm.6662
2. Steptoe A, Peacey V, Wardle J. Sleep Duration and Health in Young Adults. Arch Intern Med. 2006;166(16):1689–1692. doi:10.1001/archinte.166.16.1689
Lines 141–142: It seems likely that sleep quality, asked in this way, already incorporates some aspects of sleep duration. While it might not be possible to disentangle this here, the point should be considered in the Discussion.
Comment: Accepted. We stated in our ‘Limitation’ section that the association between sleep quality and diet should be interpreted with caution (despite cross-sectional design), since sleep quality already incorporates some aspects of sleep duration.
Lines 142–144: Note that it seems very likely that some of these variables could lie in part on the causal pathway between sleep and diet. For example, chronic sleeplessness might lead to psychological distress, which could lead to poor dietary choices. Other arguments could be made for PA and self-rated health, and perhaps even BMI and some chronic diseases.
Comment: Accepted. We performed the analyses with only confounders put into the model and found similar associations that we have now. Also, we performed an analysis where we only put sleep duration and sleep quality separately into the model and also found similar odds ratios. We stated them in the ‘Results’ section.
Lines 142–144: Why not age and sex? These seem logical predictors of and potential confounders between sleep and diet quality.
Comment: Accepted. We performed an interaction effect analysis and found no significant differences between diet and sex and diet and age and we put this in the ‘Results’ section.
Line 144: Have you considered effect modification, for example by sex? This seems quite plausible to me (that associations between sleep and diet quality could vary between men and women).
Comment: Accepted. We performed an interaction effect analysis and found no significant differences between diet and sex and diet and age and we put this in the ‘Results’ section.
Line 144: Given the issues mentioned above of potential overadjusting for variables on the causal pathway, it would be useful to show a causal model (e.g. a directed acyclic graph) so the reader can understand your view of the variables’ interrelationships.
Comment: Accepted. We performed the analyses with only confounders put into the model and found similar associations that we have now. Also we performed an analysis where we only put sleep duration and sleep quality separately into the model and also found similar odds ratios. We stated them in the ‘Results’ section.
Line 144: I would suggest that you show results from unadjusted (sleep measure only) models, partially-adjusted (for variables you are confident are not on the causal pathway) models, and fully-adjusted (for all your variables of interest) models so you can comment on possible mediation effects. At the moment, with only the fully adjusted model, the reader cannot consider issues about overadjustment.
Comment: Accepted. We performed the analyses with only confounders put into the model and found similar associations that we have now. Also we performed an analysis where we only put sleep duration and sleep quality separately into the model and also found similar odds ratios. We stated them in the ‘Results’ section.
Line 147: A standard Table 1 describing the participants and their characteristics (age, sex, ethnicity/race, etc.) is needed here.
Comment: Explained. We did that in the text at the beginning of the ‘Results’ section.
Line 160: Same issue with CI limit as in abstract.
Comment: Accepted.
Table 1: Given you use logistic regression for the adjusted models, it would be a smoother transition to also use logistic regression here rather than Chi-squared analyses. This would also allow reporting effect sizes and CIs. At the moment, it is not immediately clear to the reader what the differences between the levels of the study variables are for the diet quality outcome without working through the percentages and ORs would make this very clear immediately. You might also prefer to use row percentages when describing this data so that the percentage of respondents with low and moderate/high diet quality can be seen for each level of the study variables.
Comment: Explained. The first reviewer said that the results are clearly presented and we have seen that other authors have also presented column percentages for their results.
1. Štefan L, Juranko D, Prosoli R, Barić R, Sporiš G. Self-Reported Sleep Duration and Self-Rated Health in Young Adults. J Clin Sleep Med. 2017;13(7):899-904. Published 2017 Jul 15. doi:10.5664/jcsm.6662
2. Novak, Dario; Suzuki, Etsuji; Kawachi, Ichiro Are family, neighbourhood and school social capital associated with higher self-rated health among Croatian high school students? A population-based study // BMJ Open, 5 (2015), 6; e007184
Table 2: Actual p-values (not just asterisks) should be shown here. Note that the p-values for sleep duration (unless it is changed to a continuous variable) should be Wald tests with p-values from pairwise comparisons shown only in the text unless some form of multiplicity adjustment is made.
Comment: Accepted. We put p-values in the Table 2. Similar thing was asked by one previous reviewer.
Line 177: I’d replace “yet” with “and” here as this is not a surprising point.
Comment: Accepted.
Line 186: This isn’t the usual meaning of “sleep hygiene” as I am aware of it.
Comment: Accepted. We deleted it.
Line 197: As above.
Comment: Accepted.
Line 200: Suggest deleting “to these days” or replace with “to the present day”.
Comment: Accepted.
Lines 200–201: I think you need to describe these “protective lifestyle factors”.
Comment: Accepted.
Line 210: “effects” (add “s”)
Comment: Accepted.
Lines 210-211: FFQs are not objective.
Comment: Accepted. We deleted the FFQs sentence.
Line 211: Do you mean “with a later follow-up”? Missing confounders is perhaps an even greater threat to the validity of such a study.
Comment: Accepted. We put one sentence about the missing covariates for the future research.
Line 212: Your Discussion stops before discussing the practical and research implications of your findings. The Discussion needs to be extended to cover these.
Comment: Accepted. We extended the ‘Discussion’ section with the ‘Conclusion’.
Reviewer 3 Report
Methods
Why are there differences in Likert scale from 4 point to 5 point
Results and Differences
The results seems to be strongly stated in a cross-sectional study even though the authors suggest in the limitation section that this is self reported and hence there is recall bias and measurement error. Would use examples of those. But then to suggest that the study suggests that is obvious about long and short sleep duration and sleep quality is associated with diet is an overreach. Did they ask how many of them had untreated sleep disorder such as OSA which is common in that age group and may impact quality of sleep
Author Response
Reviewer 3
Open Review
(x) I would not like to sign my review report
( ) I would like to sign my review report
English language and style
( ) Extensive editing of English language and style required
(x) Moderate English changes required
( ) English language and style are fine/minor spell check required
( ) I don't feel qualified to judge about the English language and style
Yes | Can be improved | Must be improved | Not applicable | |
Does the introduction provide sufficient background and include all relevant references? | ( ) | ( ) | (x) | ( ) |
Is the research design appropriate? | ( ) | ( ) | (x) | ( ) |
Are the methods adequately described? | ( ) | ( ) | (x) | ( ) |
Are the results clearly presented? | ( ) | ( ) | (x) | ( ) |
Are the conclusions supported by the results? | ( ) | ( ) | (x) | ( ) |
Comments and Suggestions for Authors
Methods
Why are there differences in Likert scale from 4 point to 5 point
Comment: Accepted. We deleted the part of the sentence where we used Likert-type scale.
Results and Differences
The results seems to be strongly stated in a cross-sectional study even though the authors suggest in the limitation section that this is self reported and hence there is recall bias and measurement error. Would use examples of those. But then to suggest that the study suggests that is obvious about long and short sleep duration and sleep quality is associated with diet is an overreach. Did they ask how many of them had untreated sleep disorder such as OSA which is common in that age group and may impact quality of sleep
Comment: Accepted. We deleted the aforementioned sentence as you stated that it is overreacking and stated in our 'Limitation' section that we did not adjust for other potential covariates, such as obstructive sleep apnoea or breathing problems, which might be associated with sleep duration, sleep quality and diet.
Round 2
Reviewer 1 Report
thank you to the authors for the revised version. the revised manuscript is look more scientifically after suggested by other reviewers.
I think this paper will share the important information regarding sleep quality, duration and dietary. Congratulation
Reviewer 2 Report
Thank you to the authors for their revisions. Many things have been improved in this version.
One important point that needs to be made is that referring to other studies having done something (especially when there is overlap between the current authors and co-authors on manuscripts) does not count as explaining your decision here. The approaches used in the literature are not always correct and new practices emerge over time. You need to address the following three questions for your study without just referring to other studies as having done things in the same way as you have.
* “I would struggle to say someone who described their health as “fair” is in “good” health, this sounds more likely to indicate “poor” health to me. Comment: Explained. Previous studies have used the same categorizations”
* “Lines 134–137: … Note that Poisson regression with robust standard errors (to produce RRs with appropriate confidence intervals) would be another, more interpretable, option. See https://doi.org/10.1093/aje/kwh090 for details. Comment: Explained. Previous studies have used odds ratios for their associations and we are just referring to them.”
* “Table 1: Given you use logistic regression for the adjusted models, it would be a smoother transition to also use logistic regression here rather than Chi-squared analyses. This would also allow reporting effect sizes and CIs.…You might also prefer to use row percentages when describing this data so that the percentage of respondents with low and moderate/high diet quality can be seen for each level of the study variables. Comment: Explained. The first reviewer said that the results are clearly presented and we have seen that other authors have also presented column percentages for their results.”
For the last of these, the problem is that you interpret the results in terms of row percentages despite not presenting these. See Lines 197–199 for example: “Higher percentage of the participants with ‘optimal’ sleep duration and ‘good’ sleep quality were categorized as having ‘moderate-to-high’ healthy diet (p<0.001).” The percentages referred to here are not presented in the table due to the use of column percentages. The same also applies to the following text “Among the covariates, significantly higher percentage of participants with ‘normal’ body-mass index, ‘good’ self-rated health, ‘low’ psychological distress and ‘sufficiently’ active were categorized as having ‘moderate-to-high’ healthy diet (p<0.01).”
A few other comments from last time have not been fully addressed:
“Line 43: I don’t believe that Frank et al. make this particular point. Can you quote what you are using from their article here? Comment: Explained. Frank et al. in their abstract stated that sentence that we reworded in our manuscript, pointing out that "Dietary quality and intake of specific nutrients can impact regulatory hormonal pathways to alter sleep quantity and quality.''”
Note that you appear to be using the phrase “sleep hygiene” in a non-standard manner. This refers to habits and practices such as avoiding caffeine intake, music or audiobooks while falling asleep, physical activity prior to going to bed, etc. What you are quoting from Frank et al. arguably touches on sleep hygiene (minimising alcohol and caffeine would be the most likely overlap between the two but sleep hygiene is a much broader concept). You actually make a suggestion in the conclusion (Line 314 “reading before going to sleep”) that can be seen as going against some sleep hygiene recommendations (which would recommend that beds not be used for other activities).
“Comment: Accepted. We re-calculated the data and obtained similar values suing GEEs with Pearson deviance model.”
The text on Lines 160–161 does not describe the model itself. From what you have reported here and previously, this sounds as if you are using logistic regression (logit link and binomial errors). For GEEs, you also need to explain the type of working correlation structure used. What model diagnostics were used, if any?
“Lines 62–64: This is not a replicable calculation. What is this effect size (there are hundreds of different effect sizes), what is the level of significance, how are the number of predictors incorporated into the calculation? Note that a power calculation is not to detect the smallest change (which you cannot detect anyway in a cross-sectional study), but is intended to ensure you can detect the smallest clinically or practically important difference. Comment: Accepted. We clarified that we used f2 effect size with the power of 0.90 and we entered a total of 8 predictors into the model and this analysis led to 636 participants.”
Sorry but this is still not replicable. The reader should be able to use standard statistical software to produce this number 636 and you need to provide the information necessary for this (what is the level of significance, how is the sample being divided into groups?). The number of covariates is only really useful (beyond adjusting the degrees of freedom) if the explanatory power of those covariates is known. If you are using standardised effect sizes, you still need to explain why these are the smallest sizes that would be of interest, preferably by translating these into actual units or proportions. In any case, it looks as if you are reporting on a sample size calculation for a general linear model which is not appropriate for generalised linear models such as logistic regression, particularly when there is also clustering involved.
“Line 68: Mean height and mean weight are unlikely to be useful when reported for men and women combined. I’d suggest the new Table 1 shows variables by sex and overall. Comment: Explained. We only had 16% of men in our study and we found no significant differences between men and women in terms of diet (p=0.123), sleep duration (p=0.426) and sleep quality (p=0.295), so we dropped-out the sex-stratified analysis. We put that in our ‘Data analysis’ section.”
Still the problem remains that a mean height of 163cm does not tell me anything useful when it comprises both men and women. Similarly for the mean weight of 71.6kg. I am unable to tell if the respondents were unusually short or tall, or unusually heavy or light with this combined data. Similarly, aging affects men and women different. Even if men and women do not differ statistically in some variables, other variables do need to be broken down by sex. Again, I would strongly recommend adding sex, age, height, weight, and BMI to Table 1 with either all other variables broken down by sex or at the very least those where sexual dimorphism exists.
“Comment: Explained. Our analysis showed no significant interaction effect between diet and age (p=0.334) and we did not include age as a predictor, since we were dealing with a very homogenous sample.”
This justification does not quite work for me. Age is plausibly associated with diet and you have a mean age of 87.60 with a SD of 2.44 years, suggesting a 95% reference range of 82.8 to 92.4 which spans ages which I would not describe as homogeneous. I think you need to add age to all adjusted models.
“Lines 142–144: Why not age and sex? These seem logical predictors of and potential confounders between sleep and diet quality. Comment: Accepted. We performed an interaction effect analysis and found no significant differences between diet and sex and diet and age and we put this in the ‘Results’ section.”
Again, I think sex differences in diet are plausible enough for warrant including sex in all adjusted models. By doing this, the reader will be able to have more confidence in your results (and note that a lack of statistical significance does not mean that there is not a causal link between these variables).
“Line 147: A standard Table 1 describing the participants and their characteristics (age, sex, ethnicity/race, etc.) is needed here. Comment: Explained. We did that in the text at the beginning of the ‘Results’ section.”
I think readers will expect, as do I, that this will be included in a table rather than just in the text.
Other specific comments are:
Line 28: “attention” should not have been deleted here, just “deal of” inserted.
Line 31: These quotation marks are not needed, I was merely quoting the suggested edit.
Line 71: Delete “to”.
Lines 71–72: I’m not sure what “We set up a specific date when and where the measurement protocol will be held.” is saying.
Line 158: “tests” (add “s”)
Lines 166–168: “We found no significant differences between men and women in terms of diet (p=0.123), sleep duration (p=0.426) and sleep quality (p=0.295), so we dropped-out the sex-stratified analysis.” I would prefer this to be in the results anyway but this is not an argument against sex-stratified analyses. Such analyses would be concerned with effect modification by sex of the association and not differences between the sexes in terms of the main predictors or outcome variables.
Line 295: “obstructive” (rather than “opstructive”)
Author Response
Reviewer 2
Open Review
( ) I would not like to sign my review report
(x) I would like to sign my review report
English language and style
( ) Extensive editing of English language and style required
( ) Moderate English changes required
(x) English language and style are fine/minor spell check required
( ) I don't feel qualified to judge about the English language and style
Yes | Can be improved | Must be improved | Not applicable | |
Does the introduction provide sufficient background and include all relevant references? | (x) | ( ) | ( ) | ( ) |
Is the research design appropriate? | ( ) | (x) | ( ) | ( ) |
Are the methods adequately described? | ( ) | (x) | ( ) | ( ) |
Are the results clearly presented? | ( ) | (x) | ( ) | ( ) |
Are the conclusions supported by the results? | ( ) | (x) | ( ) | ( ) |
Comments and Suggestions for Authors
Thank you to the authors for their revisions. Many things have been improved in this version.
One important point that needs to be made is that referring to other studies having done something (especially when there is overlap between the current authors and co-authors on manuscripts) does not count as explaining your decision here. The approaches used in the literature are not always correct and new practices emerge over time. You need to address the following three questions for your study without just referring to other studies as having done things in the same way as you have.
* “I would struggle to say someone who described their health as “fair” is in “good” health, this sounds more likely to indicate “poor” health to me. Comment: Explained. Previous studies have used the same categorizations”
Comment: Accepted. E re-calculated the values when we put fair as ‘poor’ self rated health and obtained similar values, since only 30 participants reported ‘fair’ self-rated health. We re-wrote the values in table 1 and table 2.
* “Lines 134–137: … Note that Poisson regression with robust standard errors (to produce RRs with appropriate confidence intervals) would be another, more interpretable, option. See https://doi.org/10.1093/aje/kwh090 for details. Comment: Explained. Previous studies have used odds ratios for their associations and we are just referring to them.”
Comment:Accepted. We re-calculated the data and used Poisson regression analysis using generalized estimating equations with robust errors.
* “Table 1: Given you use logistic regression for the adjusted models, it would be a smoother transition to also use logistic regression here rather than Chi-squared analyses. This would also allow reporting effect sizes and CIs.…You might also prefer to use row percentages when describing this data so that the percentage of respondents with low and moderate/high diet quality can be seen for each level of the study variables. Comment: Explained. The first reviewer said that the results are clearly presented and we have seen that other authors have also presented column percentages for their results.”
Comment: Accepted. We put in the ‘Chi-square’ test odds ratio with 95% CI. Also, we left column percentages in table 1, yet used row columns in the text.
For the last of these, the problem is that you interpret the results in terms of row percentages despite not presenting these. See Lines 197–199 for example: “Higher percentage of the participants with ‘optimal’ sleep duration and ‘good’ sleep quality were categorized as having ‘moderate-to-high’ healthy diet (p<0.001).” The percentages referred to here are not presented in the table due to the use of column percentages. The same also applies to the following text “Among the covariates, significantly higher percentage of participants with ‘normal’ body-mass index, ‘good’ self-rated health, ‘low’ psychological distress and ‘sufficiently’ active were categorized as having ‘moderate-to-high’ healthy diet (p<0.01).”
A few other comments from last time have not been fully addressed:
“Line 43: I don’t believe that Frank et al. make this particular point. Can you quote what you are using from their article here? Comment: Explained. Frank et al. in their abstract stated that sentence that we reworded in our manuscript, pointing out that "Dietary quality and intake of specific nutrients can impact regulatory hormonal pathways to alter sleep quantity and quality.''”
Note that you appear to be using the phrase “sleep hygiene” in a non-standard manner. This refers to habits and practices such as avoiding caffeine intake, music or audiobooks while falling asleep, physical activity prior to going to bed, etc. What you are quoting from Frank et al. arguably touches on sleep hygiene (minimising alcohol and caffeine would be the most likely overlap between the two but sleep hygiene is a much broader concept). You actually make a suggestion in the conclusion (Line 314 “reading before going to sleep”) that can be seen as going against some sleep hygiene recommendations (which would recommend that beds not be used for other activities).
Comment: Accepted. We deleted in the conclusion section reading the book before the bed time and stated, that being in bed should be only reserved for sleep.
“Comment: Accepted. We re-calculated the data and obtained similar values suing GEEs with Pearson deviance model.”
The text on Lines 160–161 does not describe the model itself. From what you have reported here and previously, this sounds as if you are using logistic regression (logit link and binomial errors). For GEEs, you also need to explain the type of working correlation structure used. What model diagnostics were used, if any?
Comment: Accepted. We provide more detailed information about the analysis.
“Lines 62–64: This is not a replicable calculation. What is this effect size (there are hundreds of different effect sizes), what is the level of significance, how are the number of predictors incorporated into the calculation? Note that a power calculation is not to detect the smallest change (which you cannot detect anyway in a cross-sectional study), but is intended to ensure you can detect the smallest clinically or practically important difference. Comment: Accepted. We clarified that we used f2 effect size with the power of 0.90 and we entered a total of 8 predictors into the model and this analysis led to 636 participants.”
Sorry but this is still not replicable. The reader should be able to use standard statistical software to produce this number 636 and you need to provide the information necessary for this (what is the level of significance, how is the sample being divided into groups?). The number of covariates is only really useful (beyond adjusting the degrees of freedom) if the explanatory power of those covariates is known. If you are using standardised effect sizes, you still need to explain why these are the smallest sizes that would be of interest, preferably by translating these into actual units or proportions. In any case, it looks as if you are reporting on a sample size calculation for a general linear model which is not appropriate for generalised linear models such as logistic regression, particularly when there is also clustering involved.
Comment: Accepted. We re-wrote the sample size calculation for generalized linear models.
“Line 68: Mean height and mean weight are unlikely to be useful when reported for men and women combined. I’d suggest the new Table 1 shows variables by sex and overall. Comment: Explained. We only had 16% of men in our study and we found no significant differences between men and women in terms of diet (p=0.123), sleep duration (p=0.426) and sleep quality (p=0.295), so we dropped-out the sex-stratified analysis. We put that in our ‘Data analysis’ section.”
Still the problem remains that a mean height of 4075px does not tell me anything useful when it comprises both men and women. Similarly for the mean weight of 71.6kg. I am unable to tell if the respondents were unusually short or tall, or unusually heavy or light with this combined data. Similarly, aging affects men and women different. Even if men and women do not differ statistically in some variables, other variables do need to be broken down by sex. Again, I would strongly recommend adding sex, age, height, weight, and BMI to Table 1 with either all other variables broken down by sex or at the very least those where sexual dimorphism exists.
Comment: Accepted. We added sex, age, height and weight in Table 1.
“Comment: Explained. Our analysis showed no significant interaction effect between diet and age (p=0.334) and we did not include age as a predictor, since we were dealing with a very homogenous sample.”
This justification does not quite work for me. Age is plausibly associated with diet and you have a mean age of 87.60 with a SD of 2.44 years, suggesting a 95% reference range of 82.8 to 92.4 which spans ages which I would not describe as homogeneous. I think you need to add age to all adjusted models.
Comment: Accepted. We clarified the sentence and put age in all models.
“Lines 142–144: Why not age and sex? These seem logical predictors of and potential confounders between sleep and diet quality. Comment: Accepted. We performed an interaction effect analysis and found no significant differences between diet and sex and diet and age and we put this in the ‘Results’ section.”
Again, I think sex differences in diet are plausible enough for warrant including sex in all adjusted models. By doing this, the reader will be able to have more confidence in your results (and note that a lack of statistical significance does not mean that there is not a causal link between these variables).
Comment: Accepted. We added both age and sex in adjusted models.
“Line 147: A standard Table 1 describing the participants and their characteristics (age, sex, ethnicity/race, etc.) is needed here. Comment: Explained. We did that in the text at the beginning of the ‘Results’ section.”
I think readers will expect, as do I, that this will be included in a table rather than just in the text.
Comment: Accepted. We added age, sex, height and weight in table 1.
Other specific comments are:
Line 28: “attention” should not have been deleted here, just “deal of” inserted.
Comment: Accepted.
Line 31: These quotation marks are not needed, I was merely quoting the suggested edit.
Comment: Accepted.
Line 71: Delete “to”.
Comment: Accepted.
Lines 71–72: I’m not sure what “We set up a specific date when and where the measurement protocol will be held.” is saying.
Comment: Accepted. We clarified the sentence.
Line 158: “tests” (add “s”)
Lines 166–168: “We found no significant differences between men and women in terms of diet (p=0.123), sleep duration (p=0.426) and sleep quality (p=0.295), so we dropped-out the sex-stratified analysis.” I would prefer this to be in the results anyway but this is not an argument against sex-stratified analyses. Such analyses would be concerned with effect modification by sex of the association and not differences between the sexes in terms of the main predictors or outcome variables.
Comment: Accepted. We clarified and stated that we performed an interaction effect analysis between sex and sleep duration and sleep quality and due to a non-significant effect, we dropped-out the sex-stratified analysis.
Line 295: “obstructive” (rather than “opstructive”)
Comment: Accepted.